# Cyclophilin Inhibitor Rencofilstat Combined with Proteasome Inhibitor Ixazomib Increases Proteotoxic Cell Death in Advanced Prostate Cancer Cells with Minimal Effects on Non-Cancer Cells

**DOI:** 10.3390/biomedicines13102442

**Published:** 2025-10-07

**Authors:** Carlos Perez-Stable, Alicia de las Pozas, Medhi Wangpaichitr, Robert T. Foster, Daren R. Ure

**Affiliations:** 1Research Service, Bruce W. Carter Veterans Affairs Medical Center, Miami, FL 33125, USA; delaspozasalicia@gmail.com (A.d.l.P.); mwangpaichitr@med.miami.edu (M.W.); 2South Florida VA Foundation for Research and Education, Miami, FL 33125, USA; 3Geriatric Research, Education, and Clinical Center, Bruce W. Carter Veterans Affairs Medical Center, Miami, FL 33125, USA; 4Department of Medicine, Division of Gerontology & Palliative Medicine, University of Miami Miller School of Medicine, Miami, FL 33136, USA; 5Sylvester Comprehensive Cancer Center, University of Miami Miller School of Medicine, Miami, FL 33136, USA; 6Department of Surgery, Division of Cardiothoracic Surgery, University of Miami Miller School of Medicine, Miami, FL 33136, USA; 7Hepion Pharmaceuticals, Edmonton, AB T5J 4P6, Canada; rfoster.4824@gmail.com (R.T.F.); daren.ure@gmail.com (D.R.U.)

**Keywords:** cyclophilin, proteasome, XBP1s, PERK, unfolded protein response, apoptosis

## Abstract

**Background/Objective:** Proteotoxic stress induced by inhibitors of the ubiquitin–proteasome system has been successful in multiple myeloma but not in solid cancers such as prostate cancer. Our objective is to find a combination with proteasome inhibitors that increases apoptotic cell death in all types of prostate cancer without harming non-cancer cells. **Methods:** The effectiveness of rencofilstat, a pan-cyclophilin inhibitor, combined with the ixazomib proteasome inhibitor, was investigated in multiple prostate cancer and non-cancer cells. Inducible knockdown of stress response XBP1s and cyclophilins A/B and inducible expression of XBP1s and cyclophilin B were developed in prostate cancer to determine functional roles. **Results:** Rencofilstat + ixazomib increased apoptotic cell death in prostate cancer but not in non-cancer cells. We investigated the effects on XBP1s and PERK, important unfolded protein response factors required for cells to survive proteotoxic stress. The results revealed that XBP1s had a pro-survival role early, but maintenance at later times of rencofilstat + ixazomib treatment resulted in cell death. In addition, decreased PERK and phospho-eIF2α likely maintained protein synthesis to further enhance proteotoxic stress. In contrast, rencofilstat + ixazomib did not alter XBP1s or PERK in non-cancer cells. Additional genetic experiments showed that the RCF targets cyclophilins A, B, and D had protective effects. Rencofilstat increased extracellular secretion of cyclophilin B, but rencofilstat + ixazomib reduced glycosylation and, likely, the biological function of CD147 (CypB receptor) and decreased downstream ERK signaling. **Conclusions:** Rencofilstat + ixazomib may be a new strategy for increasing proteotoxic stress and apoptotic cell death in advanced prostate cancer cells with less toxic side effects.

## 1. Introduction

Early-stage prostate cancer (PCa) responds to androgen deprivation therapy (ADT) but often develops resistance, resulting in castration-resistant PCa (CRPC) [1,2]. Recent ADT drugs (e.g., androgen receptor [AR] antagonist enzalutamide) have been successfully utilized against CRPC, but resistance eventually develops. One ADT resistance mechanism is the ability of CRPC cells to adapt by reducing AR and developing into a CRPC cell type that is not dependent on the drug target (e.g., neuroendocrine prostate cancer [NEPC]) [3,4,5]. Once late-stage advanced CRPC/NEPC resistant to current ADT strategies develops, no effective treatments are available, resulting in short survival times. However, the recent development of the PSMA-targeting radioligand ^177^Lutetium-PSMA-167 (Pluvicto) offers improvements for metastatic CRPC [6]. The identification of molecular drivers and bypass signaling pathways for NEPC has not yet led to clinical success, suggesting that new strategies are required [3,4,5].

Cancer cells must overcome the damaging effects of proteotoxic stress caused by the increased protein synthesis necessary for tumor growth. [7,8,9]. Clinical success for inhibitors of the ubiquitin–proteasome system (UPS, which degrades unfolded/misfolded proteins [10]) has been achieved for multiple myeloma but not for solid cancers such as CRPC [11,12,13]. Toxic side effects from UPS inhibitors also present a challenge for clinical use. Advanced CRPC/NEPC is less secretory compared to multiple myeloma and requires combinations with UPS inhibitors to enhance proteotoxic stress past an acceptable threshold to activate apoptotic cell death. Significantly, advanced CRPC/NEPC may be especially vulnerable to strategies that increase proteotoxic stress [14,15]. Our goal is to identify a drug that, in combination with UPS inhibitors, increases proteotoxic stress and apoptotic cell death in CRPC/NEPC, but not in normal cells.

We propose a new combination with a UPS inhibitor (ixazomib, Ixz, approved for multiple myeloma) [16,17] using the pan-cyclophilin inhibitor rencofilstat (RCF, non-immunosuppressive analog of cyclosporin A, previously known as CRV431 [18,19]) to increase apoptotic cell death in all types of PCa/CRPC/NEPC cells but with less effect on non-cancer cells. This combination originated from our unexpected finding that the addition of cyclosporin A (which blocks necrosis) to UPS inhibitors increased cell death. RCF targets the cyclophilin (Cyp) family of highly abundant proteins located in multiple cell compartments (cytosol, ER, mitochondria, and nucleus) and secretion; Cyps are important in protein folding and stress response [20,21,22]. Since CypA and B isoforms have important roles in tumor progression and stress protection, the use of Cyp inhibitors in cancer chemotherapy is warranted [23,24]. Although Cyps have been well studied in multiple biological systems [20,21,22], their functional roles in mediating drug responses in advanced CRPC/NEPC are unknown. Our hypothesis is that adding RCF to enhance misfolded proteins and reduce stress response, with Ixz to reduce protein degradation, will further amplify proteotoxic stress and apoptotic cell death. Clinical data from multiple myeloma patients resistant to UPS inhibitors support the use of this strategy [25]. In addition, we recently published encouraging results with the RCF + Ixz combination in multiple hepatocellular carcinoma cells and the Hep3B xenograft in vivo system [26]. 

The cellular pathways that respond to enhanced proteotoxic stress involve endoplasmic reticulum (ER) stress and the unfolded protein response (UPR) [27,28]. UPR is a protective mechanism against ER stress that activates the kinase signaling pathways IRE1α, PERK, and ATF6, which results in a block of protein synthesis via PERK phosphorylation of eIF2α and an increase in the ability to properly fold proteins (e.g., XBP1s-positive transcription factor for chaperones) [27,28]. Cancer cells, where the UPR is already highly activated, appear to be vulnerable to further increasing ER stress, and this may have an irreparable, toxic effect. In contrast, normal or non-cancer cells have less UPR activity and may be more resistant to proteotoxic stress. There is a significant role for ER stress and the UPR in mediating apoptotic cell death through UPS inhibitors (an increase in poly-ubiquitinated [poly-Ub] proteins) [29,30,31]. Apoptotic cell death is triggered when the UPR cannot protect against enhanced proteotoxic stress [27,28,32,33]. Since CRPC/NEPC already has a higher UPR compared to AR-sensitive PCa, it may be especially vulnerable to further increases in proteotoxic stress [14,34,35,36,37].

Here, we present data showing that the RCF + Ixz combination had a synergistic anti-proliferation effect and increased apoptotic cell death in all types of PCa/CPRC/NEPC cells. The results revealed activation of IRE1α/XBP1s and decreased PERK in response to increased ER stress. Inducible knockdown and expression experiments established the pro-survival role of XBP1s early and a pro-death role later. Furthermore, the maintenance of protein synthesis during ER stress was important for RCF + Ixz cell death, likely from decreased PERK and phospho (P)-eIF2α. Genetic experiments revealed pro-survival roles for CypA, B, and D. In non-cancer cells, RCF + Ixz did not increase cell death or XBP1s or decrease PERK/P-eIF2α. Overall, RCF + Ixz increased apoptotic cell death and amplified ER/proteotoxic stress in PCa/CPRC/NEPC cells past a tolerable threshold that the UPR could not protect, while non-cancer cells were less sensitive.

## 2. Materials and Methods

### 2.1. Chemicals

RCF was obtained from Hepion Pharmaceuticals (Edmonton, AB, Canada); Ixz (MLN2238 biologically active, MLN9708 prodrug), QVD, and panobinostat were obtained from APExBIO (Houston, TX, USA); cycloheximide, doxycycline, and polybrene were obtained from Sigma-Aldrich (St. Louis, MO, USA); cyclosporin A was obtained from Enzo Life Sciences (Farmingdale, NY, USA); FK506, bortezomib, and carfilzomib were obtained from LC Laboratories (Woburn, MA, USA); thapsigargin was obtained from Santa Cruz Biotechnology (Santa Cruz, CA, USA); and Coomassie blue, trypan blue (0.4%), and puromycin were obtained from Thermo Fisher Scientific (Waltham, MA, USA).

### 2.2. Cancer and Non-Cancer Cells 

Human AR+ androgen-sensitive (AS) PCa (LNCaP), AR+ CRPC (22Rv1), AR− CRPC (PC3, DU145), AR− NEPC (NCI-H660, LASCPC), mouse TRAMP-C2 (NEPC), and the human non-cancer cells RWPE-1 (prostate epithelial) and BJ fibroblast were obtained from the American Type Culture Collection (ATCC). All cells were used within 4 months of the original cultures. In addition, we used non-small-cell lung carcinoma H460 and H460/CR (cisplatin-resistant [38]) and melanoma A375 and A2058 (ATCC). LNCaP, 22Rv1, PC3, DU145, H460, and H460/CR cells were maintained in RPMI 1640 medium (Thermo Fisher Scientific) and 5% fetal bovine serum (R&D Systems, Minneapolis, MN, USA). H660 and LASCPC cells were maintained in Advanced DMEM/F12, B27 supplement, Glutamax (Thermo Fisher Scientific), EGF, and bFGF (R&D Systems) [39,40]. TRAMP-C2 [41] cells were maintained in EMEM (ATCC) and 10% fetal bovine serum; RWPE-1 was maintained in Keratinocyte-SFM media (Thermo Fisher Scientific); and BJ, A375, and A2058 were maintained in DMEM medium and 10% fetal bovine serum. All cells were grown with 100 U/mL penicillin, 100 μg/mL streptomycin, and 0.25 μg/mL amphotericin (Thermo Fisher Scientific).

### 2.3. Drug Treatments

Cells were cultured in media containing RCF (1–10 μM), cyclosporin A (10 μM), FK506 (10 μM), Ixz (2238 [active], 9708 [prodrug]; 15–500 nM), bortezomib (10 nM), carfilzomib (25 nM), QVD (10 μM), cycloheximide (10 μM), panobinostat (10 nM), thapsigargin (10 nM), or DMSO (0.1%) control (10 min–72 h). RWPE-1 and BJ cells were grown to confluence before initiation of treatments [42]. Floating and trypsinized attached cells were pooled for further analysis.

### 2.4. Total Cell Death Assay 

To determine the effects of the drugs on total cell death, we used the trypan blue exclusion assay, as previously described [43], from at least 2–3 independent experiments performed in triplicate. 

### 2.5. Cell Proliferation Assay and Combination Index (CI)

The CellTiter Aqueous colorimetric method from Promega (Madison, WI, USA) was used to determine cell proliferation of PCa/CRPC/NEPC cells in media containing RCF (1–10 µM), Ixz (2238 [active], 9708 [prodrug]; 10–500 nM), cyclosporin A (2–10 µM), or control (0.1% DMSO), as previously described [26]. The data were expressed as a percentage of the control from three independent experiments performed in triplicate. The CalcuSyn Version 2 software program from Biosoft (Cambridge, UK) was used to calculate the CI, where a value ≤0.7 was synergistic. 

### 2.6. Western Blot Analysis

Preparation of total protein lysates and Western blot analysis was performed as previously described [44]. The following antibodies were used: cl-PARP (9541), XBP1s (D2C1F), LC3B (2775), eIF2α (9722), phospho (P)-eIF2α (Ser51), PERK (C33E1), IRE1α (14C10), ERK1/2 (9102), and P-ERK1/2 (9101) from Cell Signaling Technology (Danvers, MA, USA); Ub (P4D1); p62 (D3), GRP78 (A10), AR (441), CD147 (8D6), CypD (G9, referred as CypF or PPIF), actin (C-11), mouse anti-rabbit IgG-HRP (2357), and m-IgG-Fc BP-HRP (525409) from Santa Cruz Biotechnology; CypB (16045) and P-PERK (T982) from Abcam (Cambridge, MA, USA); CypA (SA296) from Enzo Life Sciences (Farmingdale, NY, USA); and and P-IRE1α (Ser724) from Novus Biologicals (Centennial, CO, USA). Precision Plus Protein Dual Color Standards (Bio-Rad Laboratories, Hercules, CA) were used to estimate molecular weights in kDa. Markers were used to cut blots horizontally so that high-, medium-, and low-molecular-weight targets could be analyzed separately with the appropriate antibodies, as previously described (26). Loading controls involved Coomassie blue staining of total proteins transferred to the membrane. Protein bands were quantified using UN-SCAN-IT digitizing software version 5.1 from Silk Scientific (Provo, UT, USA). 

### 2.7. Extracellular Secretion of Cyp by RCF 

To detect extracellular secretion of CypA and CypB in RCF-treated LNCaP cells, we used the Amicon Ultra-4 Centrifugal Filter Unit Ultracel-10 (Sigma-Aldrich). Media from LNCaP cells treated with RCF (2 μM) for 24 h was added to the filter and centrifuged at 4000× *g*, and the concentrated filtrate was analyzed by Western blot. The negative control was adding media without cells to the filter. 

### 2.8. Inducible Knockdown of XBP1s, CypA, and CypB 

The shRNA design, lentivirus production, and infection were performed as previously described [45]. The development of inducible knockdowns of human XBP1s, CypA, and CypB using the TET-pLKO puro DNA lentivirus vector (Addgene, Watertown, MA, USA; 21915 [46]) was as previously described [26]. The control shRNA was TET-pLKO puro scrambled (Scr) DNA (Addgene; 47541 [47]). Initial studies used the pLKO puro DNA lentivirus vector (Addgene; [45]) to clone the same shCypA DNA oligonucleotides [48] and shCypD1 (PPIF): CCGGGTTCTTCATCTGCACCATAAACTCGAGTTTATGGTGCAGATGAAGAACTTTTTG; shCypD2: CCGGATAGAATCTTTCGGCTCTAAGCTCGAGCTTAGAGCCGAAAGATTCTATTTTTTG. The control shRNA was targeted against green fluorescent protein (pLKO.1-GFP) (Addgene [49]). 

### 2.9. Inducible Expression of XBP1s and CypB 

XBP1s and CypB mRNA were PCR-amplified (Q5 high-fidelity PCR kit E0555S; New England Biolabs, Ipswich, MA, USA) and cloned into a pCW57-RFP-P2A-MCS DNA lentivirus vector (Addgene [50]), as previously described [26]. Lentivirus production, infection of LNCaP and 22Rv1 cells, and doxycycline induction were as described above [45]. Negative controls were LNCaP/empty vector (EV) and 22Rv1/EV. 

### 2.10. Statistical Analysis 

The two-tailed Student’s *t*-test (unequal variance) was used to calculate statistical differences between drug-treated and control cells from 2–3 independent experiments performed in duplicate or triplicate, with *p* < 0.05 considered significant. 

## 3. Results

### 3.1. RCF + Ixz Increases Apoptotic Cell Death in PCa/CRPC/NEPC 

In human AS-PCa LNCaP, CRPC 22Rv1, PC3, DU145, and NEPC H660, LASCPC cell lines, we determined the ability of RCF + Ixz to induce cell death. RCF + Ixz greatly increased cell death in all cells compared to RCF and Ixz alone (Figure 1A) (Appendix A). The apoptosis inhibitor QVD reduced the RCF + Ixz increase in cell death and cleaved (cl)-PARP (marker of apoptosis), suggesting apoptosis (Figure 1) (Appendix A). Cyclosporin A (CsA) + Ixz also increased cell death in 22Rv1, but FK506 (non-Cyp binding immunosuppressive) + Ixz had no effect (Appendix A), suggesting that targeting Cyps was essential. RCF or CsA + Ixz synergistically inhibited PCa/CRPC/NEPC cell proliferation (CI < 0.7) (Appendix A. Additional results showed that NEPC cells were more sensitive to the RCF + Ixz combination (Appendix A).

Furthermore, RCF + bortezomib (Btz) or carfilzomib (Cfz) also increased apoptotic cell death in PC3, suggesting other UPS inhibitors were also effective (Appendix A). Additional cancers, including non-small-cell lung carcinoma H460 and H460/CR (cisplatin-resistant [38]) and melanoma A375 and A2058, were also sensitive to the RCF + Cfz combination (Appendix A). CsA + Btz had strong effects on PC3 cell morphology compared to CsA, Btz, and control cells (Appendix A). These results suggest the RCF + UPS inhibitor combination is a new general strategy for increasing proteotoxic stress and apoptotic cell death in cancer cells.

### 3.2. RCF + Ixz Increases the UPR and Has Variable Effects on Autophagy in PCa/CRPC/NEPC 

We identified the specific changes in proteins of the UPR and autophagy pathways resulting from RCF + Ixz treatment. UPS inhibitors also increase autophagy, a protective homeostatic system that reduces poly-Ub proteins and damaged organelles [51,52]. In LNCaP, 22Rv1, and PC3, RCF + Ixz increased poly-Ub (proteotoxic stress), XBP1s (UPR), and LC3B (autophagy) at 24 and 48 h. There were variable effects on p62 (autophagy [52]), where it was decreased in LNCaP and 22Rv1 (higher p62) and increased in PC3 (lower p62) (Figure 1B–D). There was a major decrease in CypB with RCF alone and RCF + Ixz, but there was no effect on CypA. Analysis at earlier times of LNCaP, 22Rv1, PC3, H660, and LASCPC revealed increased XBP1s (0.5–2 h) and decreased CypB (4–8 h) with RCF and RCF + Ixz, whereas increased poly-Ub (0.5–2 h) occurred early, and increased cl-PARP occurred later (2–8 h) with RCF + Ixz (Appendix A). In 22Rv1, RCF alone had an immediate early increase in P/total [T]-IRE1α, P-ERK (10 min), and XBP1s (1 h), followed by decreased CypB (4 h) (Appendix A), suggesting that pan-Cyp inhibition had strong effects on multiple signaling pathways. Therefore, increased proteotoxic stress, UPR, and changes in autophagy with RCF + Ixz treatment were associated with induction of apoptotic cell death in multiple PCa/CRPC/NEPC cells. 

### 3.3. RCF Increases Extracellular Secretion of CypB and RCF + Ixz Reduces Glycosylated CD147 (Receptor for Secreted Cyp) and ERK Signaling

Extracellular secretion of Cyps resulting from RCF (or CsA) is not likely to be a concern because RCF will bind and inhibit Cyps regardless of location. An important pro-tumorigenic characteristic of CypA and CypB is extracellular secretion from the cancer cell and binding to the CD147 receptors (immunoglobulin family) of nearby cancer cells to increase multiple signaling pathways [21,53]. This results in an increase in proliferation, metastases, drug resistance, and survival of cancer cells. In LNCaP cells, treatment with RCF increased secretion of CypB but not CypA (Figure 2A). Further analysis indicated that RCF + Ixz reduced the glycosylation of CD147 better than RCF and Ixz alone (Figure 2B). A reduction in CD147 glycosylation has been shown to reduce its biological function [54,55]. A possible downstream consequence of the reduced CD147 function caused by RCF + Ixz was decreased P-ERK (Figure 2B). The same protein lysates showed the expected differences in cl-PARP, poly-Ub, P/T-IRE1α, XBP1s, and LC3B resulting from RCF + Ixz (Appendix A). These results suggested that, in addition to promoting apoptotic cell death in cancer cells, RCF + Ixz may reduce the pro-tumorigenic effects of extracellularly secreted Cyp and CD147 function to further improve efficacy. 

### 3.4. Maintaining Protein Synthesis Is Required for RCF + Ixz-Induced Cell Death

Addition of cycloheximide (Chx), a protein synthesis inhibitor, to RCF + Ixz decreased cell death in 22Rv1 and PC3 (Figure 3A). Further results showed that Chx + RCF + Ixz decreased cl-PARP, poly-Ub, XBP1s (24 h), and LC3B compared to RCF + Ixz, but there were no changes in CypA or CypB (Figure 3B). In order to better survive ER stress, cells shut down protein synthesis via the PERK/eIF2α pathway (ER stress activates PERK kinase to increase P-eIF2α and block initiation of translation) [27,28]. RCF + Ixz decreased total (T)-PERK and P-eIF2α (slight decrease in T-eIF2α), suggesting this protection mechanism against ER stress was blocked (Figure 3C). Therefore, it is likely that RCF + Ixz maintenance of protein synthesis under ER stress further enhances proteotoxic stress and cell death in CRPC. 

### 3.5. Lack of Toxic Effects of RCF + Ixz on Non-Cancer Cells

To determine the effects of RCF + Ixz on non-cancer cells, we used RWPE-1 (prostate epithelial) and BJ fibroblast cells. The results indicated RCF + Ixz induced much less cell death in RWPE-1 (3.2%) and BJ (6.0%) compared to H660 (96%) and 22Rv1 (49%) (Figure 4A) (Appendix A). Unlike in CRPC, RCF + Ixz did not increase cl-PARP, XBP1s, or LC3B in RWPE-1 cells, whereas in BJ cells, there was increased cl-PARP (despite low cell death) (Figure 4B) (Appendix A). However, non-cancer cells were not affected by RCF + Ixz’s increase in poly-Ub, suggesting they are more resistant to proteotoxic stress. In addition, the decrease in CypB caused by RCF + Ixz (and RCF alone) also occurred in RWPE-1, whereas there was no clear effect on LC3B (despite increased p62, suggesting inhibition of autophagy). In contrast to CRPC, RCF + Ixz did not decrease PERK or P-eIF2α in RWPE-1 cells (Figure 4C). Therefore, RCF + Ixz may be non-toxic in non-cancer cells due to fewer effects on cell death, apoptosis, XBP1s, and PERK/P-eIF2α. 

In BJ cells, increasing Ixz in the RCF + Ixz combination resulted in increased cell death (Appendix A). We suggest increasing the concentration of RCF permits the use of lower doses of Ixz (less toxic to normal cells) without affecting efficacy against CRPC cells. In contrast, using the clinically relevant pan-histone deacetylase (HDAC) inhibitor panobinostat + Ixz enhanced cell death in both 22Rv1 (71%) and BJ (66%) (Appendix A). 

A comparison of CypA and CypB protein levels revealed that (1) CypA is 2.8-fold higher in PCa/CRPC/NEPC compared to RWPE-1 non-cancer cells, and (2) CypB is 2-fold lower in PC3, H660, and LASCPC compared to LNCaP, 22Rv1, and RWPE-1 (Appendix A). Although correlative, these results provide support for increased CypA in cancer vs. non-cancer cells and a possible explanation for why PC3/NEPC cells are more sensitive to RCF + Ixz, i.e., lower CypB. 

### 3.6. XBP1s Is Pro-Survival Early and Pro-Death Later 

To establish the functional roles for XBP1s, CypA, and CypB in RCF + Ixz cell death in 22Rv1 and LNCaP, we created a doxycycline (Dox)-inducible shRNA knockdown system using lentivirus (Figure 5A) (Appendix A). The inducible (+Dox) knockdown values compared to a −Dox control (fold decrease) are 4–9 (XBP1s), 6–14 (CypA), and 7–100 (CypB). XBP1s is commonly considered a pro-survival UPR factor; however, there is evidence suggesting that with the maintenance of XBP1s, it shifts to a pro-death factor [56,57,58,59,60,61,62,63]. Dox-inducible knockdown of XBP1s at time 0 h of the RCF + Ixz treatment resulted in increased cell death in 22Rv1 and LNCaP at 27 h. In contrast, Dox induction of XBP1s knockdown 24 h after initiation of the RCF + Ixz treatment decreased cell death (Figure 5B) (Appendix A). There were no significant changes in RCF + Ixz cell death −/+ Dox in the negative control 22Rv1/shScr or LNCaP/shScr. These results suggest that XBP1s was initially a pro-survival factor, but with maintenance of XBP1s at later times of RCF + Ixz treatment, it was converted into a pro-death factor. 

### 3.7. RCF Targets CypA, B, and D Are Pro-Survival 

Inducible knockdown of CypA (similar to RCF effect) increased Ixz and RCF + Ixz cell death in 22Rv1 and LNCaP (Figure 5C) (Appendix A), indicating it was pro-survival. With CypB, inducible knockdown increased Ixz cell death, indicating it was also pro-survival. However, there were no differences −/+ Dox in RCF + Ixz cell death (Figure 5D) (Appendix A), likely due to CypB already being decreased by RCF. Early investigations using stable knockdown of CypA or CypD (also known as CypF or PPIF) revealed increased cell death in PC3 when treated with RCF + Cfz or Cfz alone (Appendix A), suggesting pro-survival roles. There were no significant changes in Ixz or RCF + Ixz cell death −/+ Dox in the negative control 22Rv1/shScr or LNCaP/shScr.

### 3.8. Additional Support for XBP1s as a Pro-Death Factor and CypB as a Pro-Survival Factor 

To further establish the functional roles of XBP1s and CypB in RCF + Ixz cell death in LNCaP and 22Rv1, we created a Dox-inducible expression system using lentivirus (Figure 6). Inducible expression of XBP1s (slight increase of 1.5-2-fold above −Dox control) resulted in greater RCF + Ixz cell death in LNCaP and 22Rv1 (Figure 6A), consistent with XBP1s promoting the death of cancer cells at later times (48 h). Inducible expression of CypB (2-3-fold) resulted in less RCF + Ixz cell death in LNCaP and 22Rv1 (Figure 6B), consistent with CypB supporting the survival of cancer cells. There were no significant changes in RCF + Ixz cell death or in expression of XBP1s or CypB −/+ Dox in negative control LNCaP/EV and 22Rv1/EV. 

## 4. Discussion

Due to the rise in the use of potent AR antagonists such as enzalutamide, there has been an increase in the development of advanced CRPC/NEPC resistant to ADT, with limited treatment options and poor patient prognosis [1,2,3,4,5]. We identified a new combination strategy for treating all types of PCa/CRPC/NEPC cells with RCF + Ixz, which increases ER and proteotoxic stress above a tolerable threshold that the UPR system cannot protect, leading to apoptotic cell death. Significantly, RCF + Ixz treatment exposes a vulnerability to elevated levels of ER and proteotoxic stress in PCa/CRPC/NEPC (and other cancers) in comparison to normal cells/tissues. Therefore, this begins to address one of the limitations of finding new and effective drugs and combinations that have fewer toxic side effects. The RCF + Ixz or Cfz combination is effective (no inherent resistance) in all cancer cell lines we have evaluated (a total of 14, including hepatocellular carcinoma [26], non-small cell lung carcinoma, and melanoma; broader testing is required), which should be an objective in experimental therapeutics research. Our data revealed that more advanced CRPC/NEPC cells (without AR) were more sensitive to RCF + Ixz. Further support for this strategy is the identification of CypA as a resistance gene to proteasome inhibition in relapsed multiple myeloma patients, and the finding that the addition of CsA to a proteasome inhibitor can overcome this resistance [25]. 

A possible explanation for the RCF + Ixz pro-death effect is the ability to maintain the elevated level of the UPR protein XBP1s compared to RCF alone at later times when apoptosis occurs in PCa/CRPC/NEPC cells. Using inducible knockdown and expression systems, we showed that maintenance of XBP1s at later times was important in increasing RCF + Ixz cell death in PCa/CRPC/NEPC. However, in non-cancer cells, RCF + Ixz did not increase XBP1s, and this was correlated with a lack of cell death. XBP1s is considered to be an important UPR pro-survival transcription factor required in response to increased unfolded/misfolded proteins resulting from ER/proteotoxic stress. However, if cells cannot overcome excessive ER/proteotoxic stress, maintenance of XBP1s at later times converts it into a pro-death factor [56,57,58,59,60,61,62,63]. One study suggests that sustained XBP1s transcriptionally activates KLF9, which enhances proteotoxic stress by activating genes important for calcium release from the ER, resulting in cell death [59]. It is likely that the expression of XBP1s in stressed non-cancer cells requires a strong mechanism to keep it at low levels in order to prevent death. Overall, further investigations are required to investigate the idea that XBP1s is an important mediator of drug responses. 

There is evidence that the PERK UPR pathway is selectively activated in advanced PCa, resulting in elevated P-eIF2α, and changes in protein synthesis that promote aggressive tumor development, and this provides a biomarker for poor patient survival [14]. In addition, ADT-resistant CRPC/NEPC (low or no AR) increases the eIF4F translation initiation complex to promote tumor cell proliferation, which provides a druggable nexus [15,64]. These studies suggest that advanced CRPC/NEPC may be especially vulnerable to strategies that promote proteotoxic stress. RCF + Ixz decreased PERK (a major kinase for eIF2α), leading to lower P-eIF2α, and likely resulting in protein synthesis remaining active in conditions where ER/proteotoxic stress is elevated. If protein synthesis is blocked (as with Chx), there is much less RCF + Ixz cell death, thus supporting a significant role for maintaining protein synthesis under conditions of high ER/proteotoxic stress. Therefore, the RCF + Ixz strategy for reducing PERK and preventing P-eIF2α may be especially useful in treating advanced CRPC/NEPC. Interestingly, in RWPE-1 and EA.hy926 (umbilical vein) non-cancer cells, RCF + Ixz did not decrease PERK or P-eIF2α [26]. Overall, RCF + Ixz has dual effects on the UPR system, resulting in the maintenance of XBP1s (pro-death) and blocking PERK/P-eIF2α (enhances proteotoxic stress). Thus, this may provide a unique cancer-specific cell death mechanism, which warrants further investigation. 

There is clinical promise in the currently used proteotoxic-stress-promoting strategies—HDAC orheat shock protein (HSP) 90 + UPS inhibitors—albeit with some toxicity concerns in non-cancer cells and tissues [65,66]. The RCF + Ixz combination keeps its potent anti-cancer effects on all types of PCa/CRPC/NEPC cells, but with fewer effects on non-cancer cells. In addition to a lack of toxicity in non-cancer RWPE-1 and BJ cells, similar results were obtained in non-cancer EA.hy926 (umbilical vein) cells and primary human dermal fibroblasts [26]. Furthermore, we have data from a xenograft mouse model of hepatocellular carcinoma (Hep3B) showing that the RCF + Ixz combination (both orally bioavailable) inhibits tumor volumes and final weights without causing general toxicity, thus supporting this patient-friendly strategy [26]. Data suggests that cells and tissues without specific Cyp isoforms, as in knockout mice, can survive, but cells without HSP90 or multiple HDACs cannot survive [67,68,69,70,71]. Thus, the expectation is that Cyp inhibitors will have less toxicity compared to HSP90 or HDAC inhibitors. An additional advantage is that by increasing the concentration of the less toxic RCF, Ixz can be lowered to a less toxic dose and still maintain strong anti-cancer efficacy. Further investigations are warranted to determine if RCF + Ixz will be an effective anti-cancer strategy in advanced CRPC/NEPC. 

There is a clear rationale for using Cyp inhibitors as anti-cancer agents due to the effects of CypA/B on tumor progression and stress protection [23,24]. In addition to being a potent pan-Cyp inhibitor (4-10-fold greater than CsA [18]), our data show that RCF rapidly decreased CypB intracellular protein levels and induced its secretion from cells. CypB secretion induced by Cyp inhibitors has been reported by others, and it is proposed to occur because CypB is retained in the ER through its active CsA binding site rather than an ER retention sequence [72,73]. Therefore, the addition of RCF disrupts CypB retention in the ER. Elevated extracellular CypB induced by RCF is not likely to be a concern because RCF will bind and inhibit CypB regardless of its location. There is extensive data showing that extracellular secretion of CypA and CypB binds to the CD147 receptor of other cancer cells to increase proliferation, metastasis, and tumor progression [21,53]. Our data show that RCF + Ixz reduced glycosylation of CD147 (and likely its function) and downstream ERK signaling in 22Rv1 CRPC. Therefore, another possible advantage of using RCF is blocking extracellular CypA and CypB from binding to CD147 and reducing CD147 function, adding a significant benefit to anti-cancer efficacy. 

An interesting aspect of the Cyp isoform family function is the diversity of locations throughout the cell, including in the inner mitochondrial membrane, as with CypD. The necrotic cell death pathway is influenced by the CypD function’s regulation of mitochondrial permeability transition [74]. Since there is evidence indicating that necrosis can inhibit apoptosis, inhibition of CypD function and necrosis with RCF + Ixz should not be a concern [75,76,77,78]. Our data for PC3 support the pro-survival role for CypD in mediating RCF + Cfz or Cfz alone. Additional evidence in several models is supportive of using RCF and other Cyp inhibitors against hepatocellular carcinoma [18,19,22,26,79,80,81]. We suggest that the RCF + Ixz combination has promise as an anti-cancer strategy for a variety of other cancers, including PCa/CRPC/NEPC.

The limitations of this study are a lack of (1) testing RCF + Ixz in mouse models of CRPC/NEPC and (2) identifying specific information on how elevated XBP1s and reduced PERK can increase RCF + Ixz cell death in advanced CRPC/NEPC, but with fewer effects on non-cancer cells. Future studies will (1) address the AR+/ERG+ PCa (representing 40–50% of routine clinical cases [82]) cell line (e.g., VCaP) in the effectiveness of RCF + Ixz; (2) use several CRPC/NEPC models (e.g., 22Rv1, PC3, H660, or patient-derived xenografts) and patient organoids to determine the effectiveness of RCF + Ixz; (3) use genomic and proteomic tools to identify the genes and proteins that are most greatly expressed or reduced by RCF + Ixz; (4) determine if expression of XBP1s in non-cancer cells is lethal; (5) investigate the role of PERK in mediating sensitivity to RCF + Ixz; and (6) determine if acquired resistance can be obtained with long-term RCF + Ixz treatment. 

## 5. Conclusions

The identification of new and effective treatments for advanced CRPC/NEPC is an imperative research topic. The RCF + Ixz strategy exposed a weakness in the ability of PCa/CRPC/NEPC (and other cancers) to survive excessive proteotoxic stress. This may be due to the unique effects of RCF + Ixz on the UPR system in maintaining XBP1s and protein synthesis (by decreasing PERK/P-eIF2α), resulting in increased cell death. Significantly, this may be an important cancer-specific mechanism because RCF + Ixz did not increase XBP1s or decrease PERK/P-eIF2α in non-cancer cells, resulting in no cell death or toxicity (Figure 7A). In addition, the ability of RCF to bind extracellular Cyp and RCF + Ixz to potentially reduce the function of CD147 (the receptor for extracellular Cyp) provides an additional therapeutic benefit (Figure 7B). Both RCF and Ixz are in clinical trials, making the clinical transition of this strategy less complicated [17,83,84]. Furthermore, the concentrations of RCF and Ixz used in this study correspond well to clinically obtained plasma concentrations [83,85]. We further emphasize the potential clinical advantage of RCF over the current HDAC or HSP90 inhibitors due to fewer toxic side effects. Future translational application of RCF + Ixz will require collaborative academic and industry efforts.

## Figures and Tables

**Figure 1 biomedicines-13-02442-f001:**
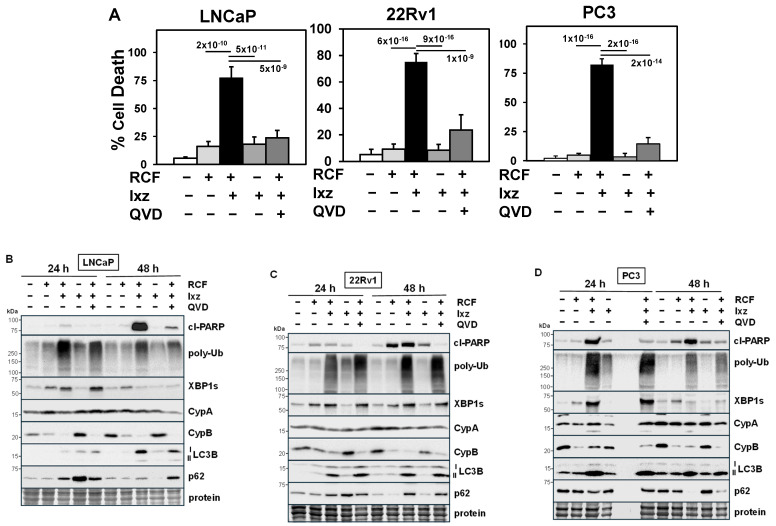
RCF + Ixz increases apoptotic cell death in PCa/CRPC cells. (**A**) Increased cell death in RCF (10 μM) + Ixz (2238, active form) (25 nM; 50 nM in LNCaP)-treated LNCaP, 22Rv1, and PC3 compared to RCF-, Ixz-, and control-treated cells (trypan blue assay; 72 h). QVD (10 μM; apoptosis inhibitor) decreased RCF + Ixz cell death in LNCaP, 22Rv1, and PC3. *p*-Values are near the bars. (**B**–**D**) Increased cl-PARP, poly-Ub, XBP1s, and LC3B in RCF + Ixz-treated (**B**) LNCaP, (**C**) 22Rv1, and (**D**) PC3 (24, 48 h) compared to RCF-, Ixz-, and control-treated cells (Western blot). CypB decreased with RCF and RCF + Ixz (24, 48 h), whereas there were variable effects on p62. No changes were observed in CypA. QVD decreased cl-PARP in RCF + Ixz. Molecular weight markers in kDa are shown to the left. Protein refers to loading control (Coomassie blue) after completion of analysis.

**Figure 2 biomedicines-13-02442-f002:**
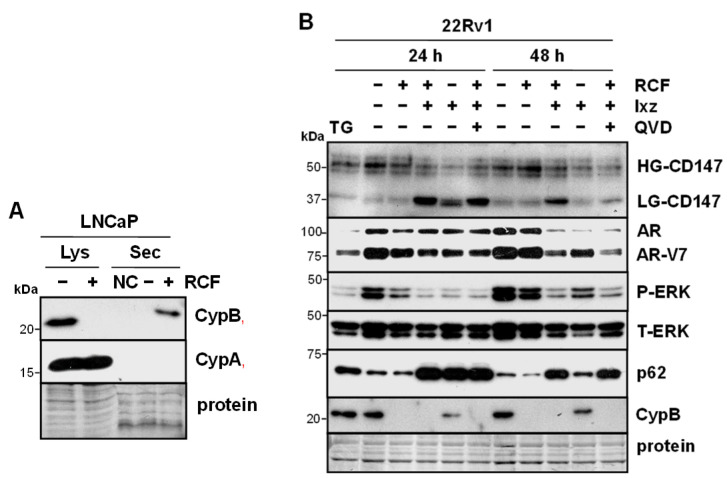
RCF increases secretion of CypB, and RCF + Ixz alters glycosylation of CD147. (**A**) Treatment of LNCaP with RCF (2 μM) for 24 h increased extracellular secretion (Sec) of CypB but not CypA (Western blot). Negative control (NC) was the addition of media without cells to the filter. Total lysate (Lys) is also shown, demonstrating a lack of CypB +RCF and no changes in CypA. (**B**) RCF (2 μM) + Ixz (9708, prodrug form) (100 nM) increased low-glycosylation (LG) CD147 (receptor for CypB) more than RCF-, Ixz-, and control-treated cells; reduction in high-glycosylation (HG) CD147 was similar in RCF + Ixz and Ixz (Western blot). Also shown are changes in AR/AR-V7, P/T-ERK (small decrease at 48 h), p62, and CypB. ER stressor thapsigargin (TG) or apoptosis inhibitor QVD had no effect on CD147. Sizes of molecular weight markers in kDa are shown to the left. Protein refers to loading control (Coomassie blue) after completion of analysis.

**Figure 3 biomedicines-13-02442-f003:**
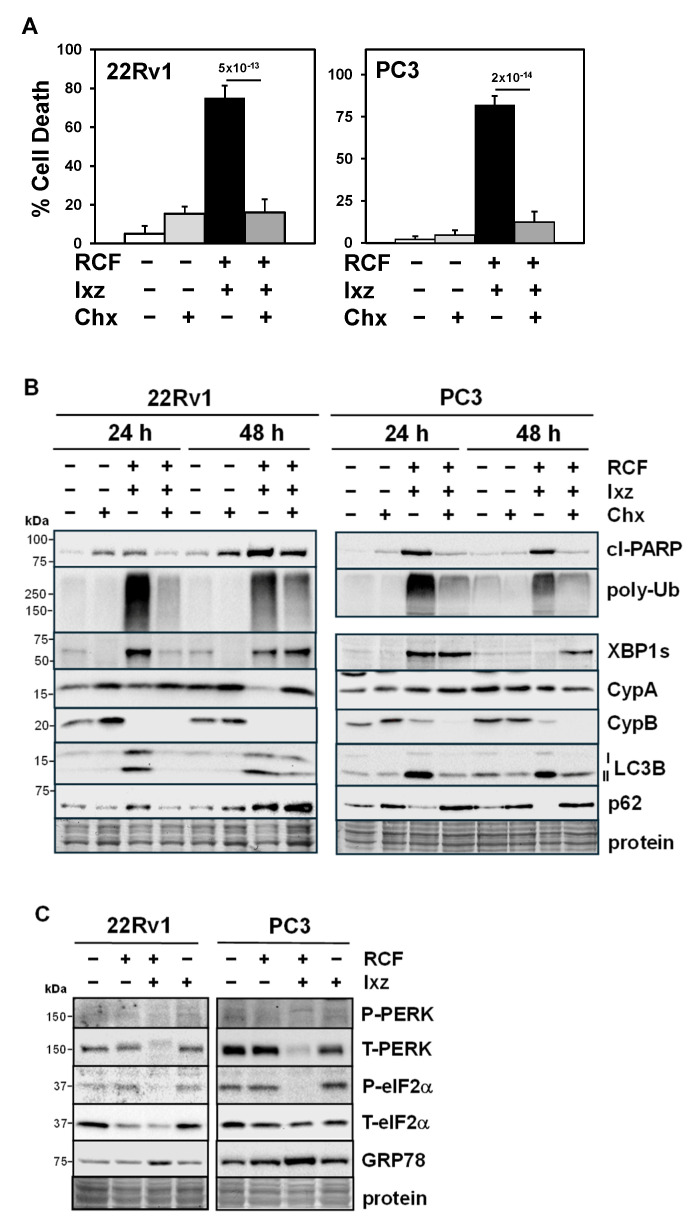
Maintaining protein synthesis is important for RCF + Ixz cell death in CRPC cells. (**A**) Chx (10 μM) + RCF (10 μM) + Ixz (2238, active form) (25 nM) significantly decreased cell death in 22Rv1 and PC3 compared to RCF + Ixz (trypan blue assay; 72 h). *p*-Values are above the bars. (**B**) In 22Rv1 and PC3, Chx + RCF + Ixz decreased cl-PARP, poly-Ub, and LC3B compared to RCF + Ixz (Western blot). Chx decreased RCF + Ixz’s effect on XBP1s in 22Rv1 (24 h) but not in PC3, and the effects on p62 were reduced (decreased in 22Rv1 and increased in PC3). There were no changes in CypA or B. (**C**) In 22Rv1 (24 h) and PC3 (48 h), RCF + Ixz decreased T-PERK and P-eIF2α compared to RCF, Ixz, and control (Western blot). RCF + Ixz increased GRP78, a marker of ER stress. Molecular weight markers in kDa are shown to the left. Protein refers to loading control (Coomassie blue) after completion of analysis.

**Figure 4 biomedicines-13-02442-f004:**
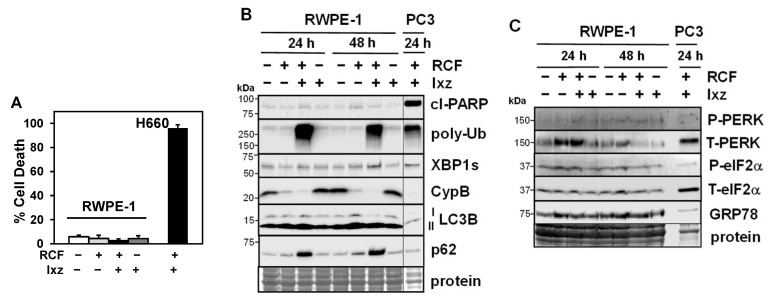
RCF + Ixz does not affect cell death in non-cancer RWPE-1 prostate epithelial cells. (**A**) RCF (5 μM) + Ixz (2238, active form) (25 nM) did not increase cell death in RWPE-1 cells (3.2%) compared to RCF-, Ixz-, and control-treated cells, but it greatly increased cell death in H660 NEPC cells (96%) (trypan blue assay; 72 h). (**B**) RCF + Ixz had minimal effects on cl-PARP and XBP1s, unlike the greater increases in PC3 (Western blot). There was a similar increase in poly-Ub in both RWPE-1 and PC3. P62 was greatly increased in RWPE-1, suggesting inhibition of autophagy. There were no changes in CypA/B and LC3B. Vertical line refers to the removal of empty lanes between RWPE-1 and PC3 samples from the same blot. (**C**) Unlike in PCa cells, RCF + Ixz had no effects on PERK or P-eIF2α in RWPE-1 cells (Western blot). No clear difference in GRP78 was noted. Molecular weight markers in kDa are shown to the left. Protein refers to loading control (Coomassie blue) after completion of analysis.

**Figure 5 biomedicines-13-02442-f005:**
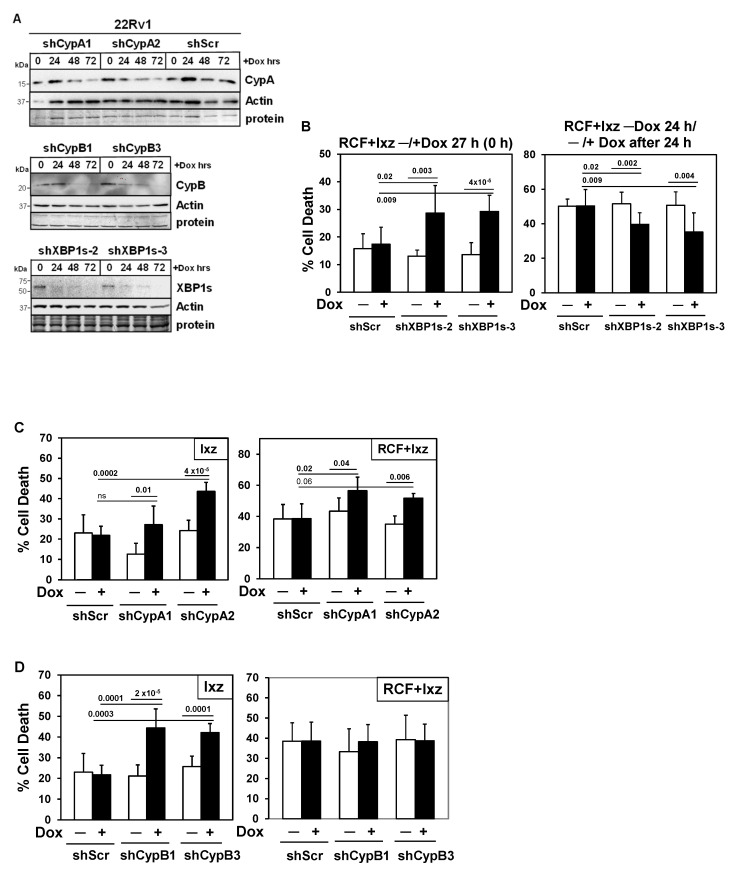
Inducible knockdowns: Determining the functional roles of XBP1s and CypA/B in 22Rv1 cells treated with Ixz or RCF + Ixz. (**A**) Dox treatment (100 ng/mL; 24, 48, 72 h) of 22Rv1/shCypA-1, -2, shScr (top blot), 22Rv1/shCypB-1, -3 (middle blot), and 22Rv1/shXBP1s-2, -3 (bottom blot) indicated less CypA (top), CypB (middle), and XBP1s (bottom) compared to control (0 h) and actin. Negative control 22Rv1/shScr showed no differences + Dox in CypA (top blot). Molecular weight markers in kDa are shown to the left. Protein refers to loading control (Coomassie blue) after completion of analysis. (**B**) Dox (+) knockdown of XBP1s starting at 0 h increased cell death in RCF (10 μM) + Ixz (2238, active form) (25 nM)-treated 22Rv1/shXBP1s-2 and -3 (27 h) (trypan blue assay). When Dox was added 24 h after initiation of RCF + Ixz, cell death was inhibited after another 24 h (48 h total). (**C**) Dox (+) knockdown of CypA increased Ixz (72 h) and RCF + Ixz (48 h) cell death in 22Rv1/shCypA-1 and -2 compared to no Dox (−). (**D**) Dox (+) knockdown of CypB increased cell death in Ixz but not in RCF + Ixz-treated 22Rv1/shCypB-1 and -3. Negative control 22Rv1/shScr showed no differences in +Dox compared to –Dox. Cells were treated with Dox for 48 h before starting the RCF + Ixz 27 h treatment. *p*-Values are above the bars.

**Figure 6 biomedicines-13-02442-f006:**
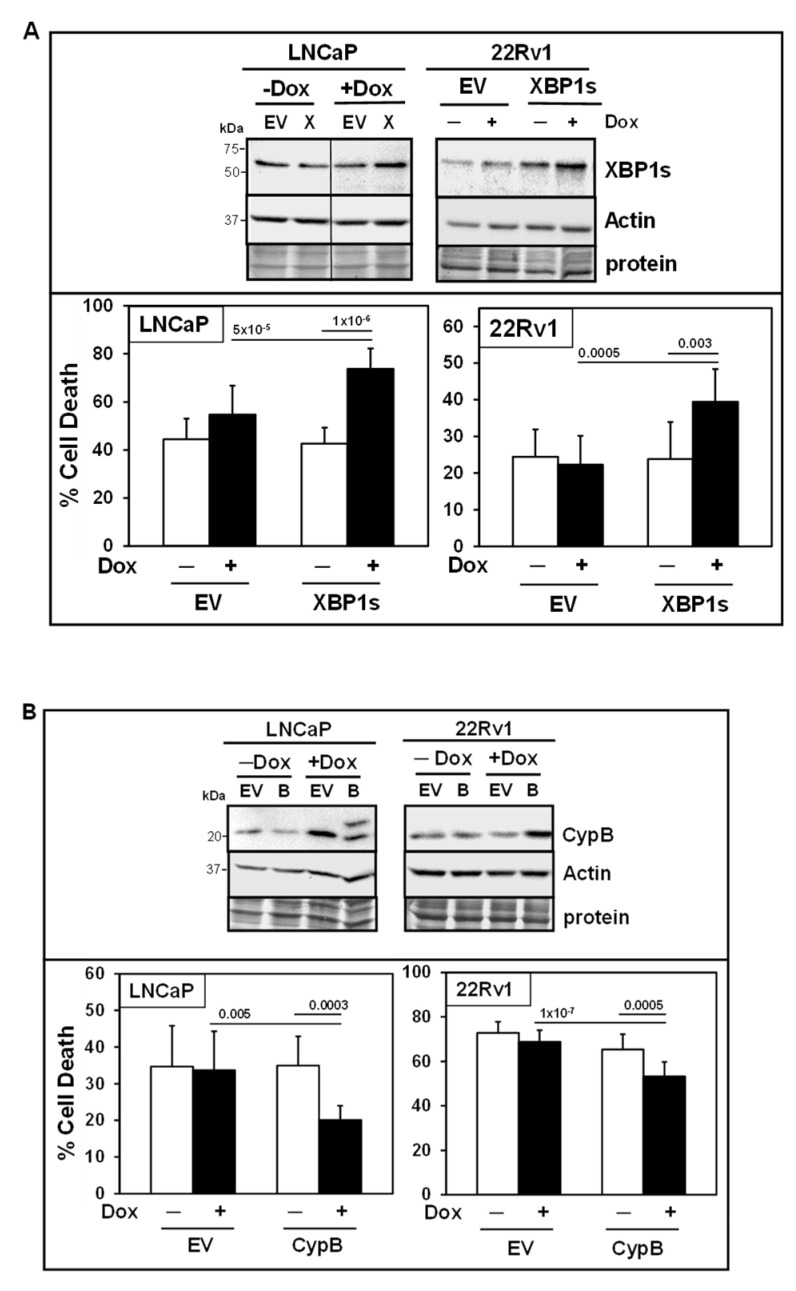
Inducible expression: Further support for XBP1s and CypB in mediating RCF + Ixz cell death. (**A**) Top panel: Dox (+) slightly increased XBP1s but not actin in LNCaP/XBP1s (X) and 22Rv1/XBP1s treated with RCF (10 μM) + Ixz (2238, active form) (25 nM) for 48 h compared to LNCaP/EV and 22Rv1/EV negative control and no Dox (−). The vertical line in the LNCaP blot refers to removal of lanes between samples from the same blot for clarity of presentation. Bottom panel: Dox (+) induction of XBP1s increased cell death in LNCaP/XBP1s and 22Rv1/XBP1s treated with RCF + Ixz compared to EV negative controls and no Dox (**−**) (trypan blue; 48 h). (**B**) Top panel: Dox (+) increased CypB but not actin in LNCaP/CypB (B) (48 h) and 22Rv1/CypB (72 h) treated with RCF + Ixz compared to EV negative controls and no Dox (**−**). Bottom panel: Dox (+) induction of CypB decreased cell death in LNCaP/CypB and 22Rv1/CypB compared to EV negative controls and no Dox (**−**). *p*-Values are above the bars.

**Figure 7 biomedicines-13-02442-f007:**
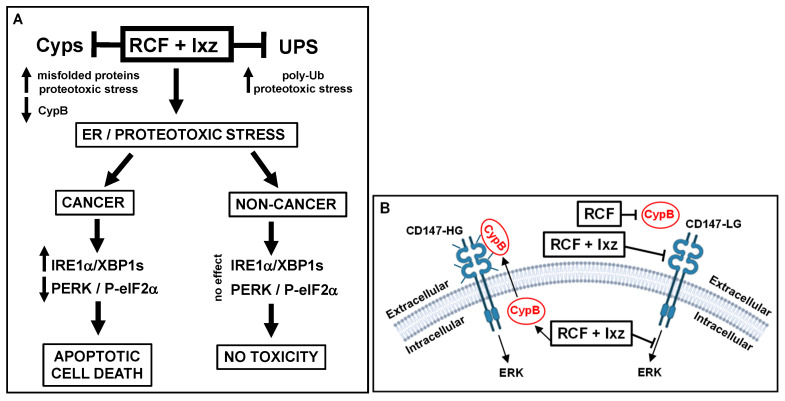
Schematic diagrams of how RCF + Ixz affects advanced CRPC/NEPC intracellular and extracellular events. (**A**) Intracellularly, RCF (pan-Cyp inhibitor) + Ixz (UPS inhibitor; increases poly-Ub) enhances proteotoxic stress. In cancer cells, RCF + Ixz activation of the IRE1α/XBP1s UPR pathway and maintenance of XBP1s result in cell death. In addition, RCF + Ixz decreases the PERK UPR pathway to block P-eIF2α (inhibits protein translation) and thus maintains protein synthesis to further enhance proteotoxic stress. In non-cancer cells, RCF + Ixz has no effect on XBP1s and PERK, resulting in no cell death or toxicity. RCF + Ixz decreases CypB and increases poly-Ub in both cancer and non-cancer cells. (**B**) RCF increases the extracellular secretion of CypB, which likely binds to the CD147-HG receptor and activates ERK signaling. RCF binds to extracellular CypB (likely decreases binding to CD147), and RCF + Ixz reduces CD147 from HG to LG, possibly reducing its function. RCF + Ixz also reduces intracellular ERK signaling. Glycosylation on the CD147 receptor is represented by small bars. LG, low glycosylation; HG, high glycosylation; 
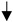
, decreased; 
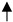
, increased.

## Data Availability

All data generated or analyzed during this study are included in this published article and Appendix A. Upon written or e-mail request, any resources or data will be made freely available.

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
