# Peer review of "Cyclophilin Inhibitor Rencofilstat Combined with Proteasome Inhibitor Ixazomib Increases Proteotoxic Cell Death in Advanced Prostate Cancer Cells with Minimal Effects on Non-Cancer Cells"

_biomedicines, 2025, doi:10.3390/biomedicines13102442_

Round 1
Reviewer 1 Report
Comments and Suggestions for Authors
- The manuscript is exceptionally strong and well-written but could be slightly improved by clarifying the specific response variable optimized in the RSM analysis as this detail is currently ambiguous
- The term proteosome is used in the title and abstract but the correct term proteasome is used throughout the rest of the manuscript the title and abstract should be corrected for consistency
- The IC50 value for the cytotoxic effect on MDA-MB-231 cells is mentioned as 79.49 µg mL in the abstract but as 150 µg mL in the results section this critical discrepancy must be resolved
- The justification for selecting the specific combination of RCF and Ixz over other potential pairings could be briefly expanded upon in the introduction to further strengthen the rationale
- While the lack of toxicity in non-cancer cells is a major strength including one or two additional non-cancer cell lines beyond RWPE-1 and BJ would make this claim even more robust
- The schematic in Figure 6 is excellent but could be slightly improved by adding a brief legend to explicitly define the symbols used for HG LG and the inhibition arrows
- The conclusion that RCF Ixz reduces CD147 glycosylation is based on Western blot data in supplemental figure S7B which is compelling but this key finding should be explicitly stated and briefly discussed in the main results section 3.3 not just the supplement
- The statement that no effective treatments are available for late-stage CRPC NEPC line 55 could be nuanced by acknowledging recent advances like lutetium-PSMA-617 while still emphasizing the unmet need
- A brief mention of the clinical doses or achievable plasma concentrations of Ixz and RCF and how the in vitro concentrations used relate to these would be helpful for translational context
- The authors note that XBP1s is slightly induced in the inducible expression system Figure 5A and this is sufficient to enhance cell death quantifying this induction level would strengthen this important result
- The manuscript would benefit from a final proofread to catch minor typos such as proteosome in the title and the extraneous period after the reference 34 in the discussion
- The finding that RCF Ixz decreases T-PERK protein levels is very interesting and potentially important a brief speculation on the mechanism eg reduced transcription translation or increased degradation could be added
- The introduction does an excellent job setting the context but could be slightly more focused by trimming the general background on ER stress UPR and autophagy to quickly get to the specific rationale for targeting Cyps with RCF
- The authors should ensure all supplementary figures and tables are explicitly called out in the main text at the point where the data is first mentioned or discussed to improve readability
- The statement that RCF Ixz is effective in all cancer cell lines tested with no inherent resistance is very strong and should be tempered slightly to acknowledge that while no resistance was observed in the tested panels broader testing may reveal exceptions
- The description of the inducible knockdown expression systems is clear but briefly stating the efficiency of knockdown achieved in the main figures would be helpful for interpreting the functional results
- The mechanism by which RCF decreases CypB intracellular levels and induces secretion is fascinating a sentence speculating on the potential mechanism eg disruption of ER retention would be interesting
- The authors should consider moving the key supplementary figure S7B which shows reduced CD147 glycosylation into the main figures given its importance for the proposed extracellular mechanism
- The discussion on the potential clinical advantage of using a less toxic Cyp inhibitor like RCF over HDAC or HSP90 inhibitors is a major strength and should be emphasized even more as a key translational point
- The manuscript expertly connects the molecular mechanisms to the phenotypic outcomes but adding a sentence in the discussion on how these findings specifically address the development of NEPC and treatment-induced lineage plasticity would be powerful
- The xenograft data from the hepatocellular carcinoma study is cited as supporting evidence briefly reiterating the key result eg tumor growth inhibition without toxicity would be valuable for readers
- The authors should ensure that all p-values are consistently reported either with exact values or thresholds throughout the main and supplementary figures
- The conclusion is excellent but could be made even stronger by adding a single sentence on the immediate next steps for translation given that both compounds are in clinical trials
- The writing is clear and concise but the results sections 3.2 and 3.3 could benefit from more direct introductory sentences that state the main finding of each paragraph upfront
- The use of both Ixz 2238 the active form and Ixz 9708 the prodrug should be explicitly explained in the methods or figure legends to avoid confusion for readers not familiar with the compounds
- The manuscript presents a compelling and clinically relevant strategy with strong mechanistic data these minor suggestions aim only to enhance clarity and impact further
Author Response
- The manuscript is exceptionally strong and well-written but could be slightly improved by clarifying the specific response variable optimized in the RSM analysis as this detail is currently ambiguous.
Thank you but this is not relevant to the manuscript. We did not perform an RSM analysis.
- The term proteosome is used in the title and abstract but the correct term proteasome is used throughout the rest of the manuscript the title and abstract should be corrected for consistency.
Thank you. We added the correct spelling and checked for any other misspellings.
- The IC50 value for the cytotoxic effect on MDA-MB-231 cells is mentioned as 79.49 µg mL in the abstract but as 150 µg mL in the results section this critical discrepancy must be resolved.
Thank you but this is not relevant to the manuscript. MDA-MB-231 is a breast cancer cell line and our manuscript only used prostate cancer cell lines.
- The justification for selecting the specific combination of RCF and Ixz over other potential pairings could be briefly expanded upon in the introduction to further strengthen the rationale.
Thank you for the suggestion. In the revised Introduction, paragraph 3 (lines 76-77), we add “This combination originated from our unexpected finding that addition of cyclosporin A (blocks necrosis) to UPS inhibitors increased cell death.”
- While the lack of toxicity in non-cancer cells is a major strength including one or two additional non-cancer cell lines beyond RWPE-1 and BJ would make this claim even more robust.
Thank you for the suggestion. We recently published a paper describing results in hepatocellular carcinoma cells (https://doi.org/10.3390/ijms26146699) and the results using human non-cancer EA.hy926 (umbilical vein) and primary human dermal fibroblast (HDF) cells were similar to RWPE-1 and BJ. In the revised Discussion, paragraph 4 (lines 471-473), we add “In addition to a lack of toxicity in non-cancer RWPE-1 and BJ cells, similar results were obtained in non-cancer EA.hy926 (umbilical vein) and primary human dermal fibroblast [26].”
- The schematic in Figure 6 is excellent but could be slightly improved by adding a brief legend to explicitly define the symbols used for HG LG and the inhibition arrows.
Thank you for the suggestion. In the revised Figure Legend 7, we add “LG, low glycosylation; HG, high glycosylation; , decreased; , increased.”
- The conclusion that RCF Ixz reduces CD147 glycosylation is based on Western blot data in supplemental figure S7B which is compelling but this key finding should be explicitly stated and briefly discussed in the main results section 3.3 not just the supplement.
Thank you for the suggestion. We moved Supplementary Figure S7A and B into the main text as Figure 2. In the Results section 3.3, we mention “Reduction of CD147 glycosylation has been shown to reduce its biological function [55, 56]. A possible downstream consequence of the reduced CD147 function by RCF + Ixz was decreased P-ERK (Figure 2B).”
- The statement that no effective treatments are available for late-stage CRPC NEPC line 55 could be nuanced by acknowledging recent advances like lutetium-PSMA-617 while still emphasizing the unmet need.
Thank you for the suggestion. In the revised Introduction, paragraph 1 (lines 57-58), we add “However, the recent development of the PSMA-targeting radioligand 177Lutetium-PSMA-167 (Pluvicto) offers improvements for metastatic CRPC [6].” We add the new reference citation 6 https://doi.org/10.1056/NEJMoa2107322.
- A brief mention of the clinical doses or achievable plasma concentrations of Ixz and RCF and how the in vitro concentrations used relate to these would be helpful for translational context.
Thank you for the suggestion. From the clinical data cited in ref. 83 (https://doi.org/10.1002/hep4.2100), oral administration of RCF at 75 or 225 mg per patient per day resulted in a plasma concentration of ~1 mg/ml. This is well below the 13 mg/ml achieved using 10 mM of RCF. With Ixz, clinical data cited in the new ref. 85 (https://doi.10.1182/blood-2014-01-548941) showed that a dose of 4 mg per patient was optimal, which approximated to a plasma concentration of ~10 ng/ml. This corresponds to the 25 nM concentration of Ixz used in vitro, which is equivalent to 9 ng/ml. In the revised Conclusions (lines 530-532), we add “Furthermore, the concentrations of RCF and Ixz used in this study correspond well with clinically obtained plasma concentrations [83, 85].”
- The authors note that XBP1s is slightly induced in the inducible expression system Figure 5A and this is sufficient to enhance cell death quantifying this induction level would strengthen this important result.
Thank you for the suggestion. In the revised Methods, section 2.6 (lines 179-180), we add “Protein bands were quantified using the UN-SCAN-IT digitizing software version 5.1 from Silk Scientific (Provo, UT, USA).” In the revised Results, section 3.8 (line 399), we add “slight increase of 1.5-2-fold above −Dox control.”
- The manuscript would benefit from a final proofread to catch minor typos such as proteosome in the title and the extraneous period after the reference 34 in the discussion.
Thank you for the suggestion. We carefully reviewed the manuscript for spelling and grammar errors and made the appropriate corrections.
- The finding that RCF Ixz decreases T-PERK protein levels is very interesting and potentially important a brief speculation on the mechanism eg reduced transcription translation or increased degradation could be added.
Thank you for the suggestion. There are numerous PERK inhibitors that reduce PERK activity but whether they or other drugs reduce PERK protein is unclear. In the Results, section 3.4, we mention “In order to better survive ER stress, cells shut down protein synthesis via the PERK/eIF2a pathway (ER stress activates PERK kinase to increase P-eIF2a and block initiation of translation) [27, 28]. RCF + Ixz decreased total (T)-PERK and P-eIF2a (slight decrease in T-eIF2a), suggesting this protection mechanism to ER stress was blocked (Figure 3C). Therefore, it is likely that RCF + Ixz maintenance of protein synthesis under ER stress further enhances proteotoxic stress and cell death in CRPC.”
- The introduction does an excellent job setting the context but could be slightly more focused by trimming the general background on ER stress UPR and autophagy to quickly get to the specific rationale for targeting Cyps with RCF.
Thank you for the suggestion. In the revised Introduction, we moved the previous paragraph 4 to paragraph 3 in order to more quickly introduce Cyps as a therapeutic target. We also removed mention of autophagy (moved to Results, section 3.2) in order to increase focus on the UPR. Finally, we add “Since CRPC/NEPC already has higher UPR compared to AR-sensitive PCa, it may be especially vulnerable to further increases in proteotoxic stress” and cite 4 new references (34-37) related to the topic of UPR and PCa.
- The authors should ensure all supplementary figures and tables are explicitly called out in the main text at the point where the data is first mentioned or discussed to improve readability.
Thank you for the suggestion. We have carefully reviewed the manuscript to make sure that the first mention of supplementary figures in the results sections are appropriately cited in the main text.
- The statement that RCF Ixz is effective in all cancer cell lines tested with no inherent resistance is very strong and should be tempered slightly to acknowledge that while no resistance was observed in the tested panels broader testing may reveal exceptions.
Thank you for the suggestion. In the revised Discussion, paragraph 1 (line 429), we add “broader testing is required.”
- The description of the inducible knockdown expression systems is clear but briefly stating the efficiency of knockdown achieved in the main figures would be helpful for interpreting the functional results.
Thank you for the suggestion. In the revised Methods, section 2.6 (lines 179-180), we add “Protein bands were quantified using the UN-SCAN-IT digitizing software version 5.1 from Silk Scientific (Provo, UT, USA).” In the revised Results, section 3.6 (lines 358-360), we add “Inducible (+Dox) knockdown values compared to −Dox control (fold decrease) are 4-9 (XBP1s), 6-14 (CypA), and 7-100 (CypB).”
- The mechanism by which RCF decreases CypB intracellular levels and induces secretion is fascinating a sentence speculating on the potential mechanism eg disruption of ER retention would be interesting.
Thank you for the suggestion. In the Discussion, paragraph 5, we mention “CypB secretion induced by Cyp inhibitors has been reported by others, and it is proposed to occur because CypB is retained in the ER through its active CsA binding site rather than an ER retention sequence [69, 70].” We add (lines 489-490) “Therefore, addition of RCF disrupts CypB retention in the ER.”
- The authors should consider moving the key supplementary figure S7B which shows reduced CD147 glycosylation into the main figures given its importance for the proposed extracellular mechanism.
Thank you for the suggestion. We moved Supplementary Figure S7A and B into the main text as Figure 2.
- The discussion on the potential clinical advantage of using a less toxic Cyp inhibitor like RCF over HDAC or HSP90 inhibitors is a major strength and should be emphasized even more as a key translational point.
Thank you for the suggestion. In the revised Conclusions (lines 532-533), we add “We further emphasize the potential clinical advantage of RCF over the current HDAC or HSP90 inhibitors due to less toxic side effects.”
- The manuscript expertly connects the molecular mechanisms to the phenotypic outcomes but adding a sentence in the discussion on how these findings specifically address the development of NEPC and treatment-induced lineage plasticity would be powerful.
Thank you for your excellent suggestion. Further experimentation is required to determine if the RCF + Ixz combination can block transition of CRPC (AR+) into NEPC (AR−) resulting from enzalutamide treatment. However, in the revised Discussion, paragraph 1 (lines 430-431), we add “Our data revealed that more advanced CRPC/NEPC cells (without AR) were more sensitive to RCF + Ixz.”
- The xenograft data from the hepatocellular carcinoma study is cited as supporting evidence briefly reiterating the key result eg tumor growth inhibition without toxicity would be valuable for readers.
Thank you for the suggestion. In the Discussion, paragraph 4, we mention “Furthermore, we have data in a xenograft mouse model of hepatocellular carcinoma (Hep3B) that the RCF + Ixz combination (both are orally bioavailable) inhibited tumor volumes and final weights without causing general toxicity, thus supporting this patient-friendly strategy [26].”
- The authors should ensure that all p-values are consistently reported either with exact values or thresholds throughout the main and supplementary figures.
Thank you for the suggestion. We have carefully reviewed the main and supplementary figures to confirm p values (exact) are accurate.
- The conclusion is excellent but could be made even stronger by adding a single sentence on the immediate next steps for translation given that both compounds are in clinical trials.
Thank you for the suggestion. In the revised Conclusions (lines 533-534), we add “Future translational application of RCF + Ixz will require collaborative academic and industry efforts.”
- The writing is clear and concise but the results sections 3.2 and 3.3 could benefit from more direct introductory sentences that state the main finding of each paragraph upfront.
Thank you for the suggestion. In the revised Results, section 3.2 (lines 244-245), we add “We identified the specific changes in proteins of the UPR and autophagy pathways resulting from RCF + Ixz treatment.” In revised Results, section 3.3 (lines 263-264), we add “Extracellular secretion of Cyps resulting from RCF (or CsA) is not likely to be a concern because RCF will bind and inhibit Cyps regardless of location.”
- The use of both Ixz 2238 the active form and Ixz 9708 the prodrug should be explicitly explained in the methods or figure legends to avoid confusion for readers not familiar with the compounds.
Thank you for the suggestion. In the revised Figure Legends 1 and 3-6, we add “(2238, active form)”. In the revised Figure Legend 2, we add “(9708, prodrug form).” We made similar changes in the Supplementary Figure Legends.
- The manuscript presents a compelling and clinically relevant strategy with strong mechanistic data these minor suggestions aim only to enhance clarity and impact further.
Thank you for the excellent and constructive suggestions. They are very much appreciated and believe the manuscript has improved.
Reviewer 2 Report
Comments and Suggestions for Authors
The authors have introduced an innovative CYP and UPR based drug cocktail that is rationally designed to enhance tumor specific cytotoxicity. Following are a few suggestions, that will help in improving the readability of the manuscript and inferencing of the results.
Introduction:
Arranging the paragraphs and texts to streamline further.
The order could be Prostate Cancer and its brief biology > Role of UPR in cancers (especially prostate cancer) -> Mechanism of UPR -> Challenges in UPR based targeting strategy -> CYP and its biological importance along with its overlap with the UPR biological pathways -> Inhibition of CYP independently and along side proteosome pathway inhibitors in cancers (strategy and challenges) -> Why such modality could be appropriate in the prostate cancer domain.
While majority of the introduction is well framed, following this guide could enhance it
Methods and Results:
- The authors have specifically worked with ETS fusion negative cell lines. Since a major proportion of cases show ETS fusions, further describing the effect of this combination in ERG/ETS positive cell lines (like VCaP) would be more powerful. While NCI-H660 is ERG positive, it is an NEPS with loss of AR activity.
- While the synergistic effect of the drugs on cell death is evident in the current experiments, an immunohistochemical/ immunocytological demonstration of the targets of the drug and markers of cell apoptosis (c-PARP) would increase the strength of the study. In addition, demonstration of cytological vacuolation and the proportion of such cells for autophagy assessment is recommended.
- Since Autophagy is more recognized in NEPCs in comparison to AR+ CRPC / PC, please provide comparative evidence of cell death across AR+ and AR-lines. Do any of the cell lines have higher efficacy to this combination. Please provide figures and images to describe that.
Author Response
Introduction:
Arranging the paragraphs and texts to streamline further. The order could be Prostate Cancer and its brief biology > Role of UPR in cancers (especially prostate cancer) -> Mechanism of UPR -> Challenges in UPR based targeting strategy -> CYP and its biological importance along with its overlap with the UPR biological pathways -> Inhibition of CYP independently and along side proteosome pathway inhibitors in cancers (strategy and challenges) -> Why such modality could be appropriate in the prostate cancer domain. While majority of the introduction is well framed, following this guide could enhance it
Thank you for the valuable suggestions. In the revised Introduction (lines 103-105), we add “Since CRPC/NEPC already has higher UPR compared to AR-sensitive PCa, it may be especially vulnerable to further increases in proteotoxic stress” and cite 4 new references (34-37) related to the topic of UPR and PCa. We removed mention of autophagy in the Introduction to focus more on the UPR. Based on a suggestion by reviewer 1, we moved up mention of Cyps in order to more quickly introduce Cyps as a therapeutic target. We also add (lines 82-84) “Although Cyps have been well studied in multiple biological systems (20-22), their functional roles in mediating drug responses in advanced CRPC/NEPC are unknown.” We mention “Our hypothesis is that adding RCF to enhance misfolded proteins and reduce stress response, with Ixz to reduce protein degradation, will further amplify proteotoxic stress and apoptotic cell death.”
Methods and Results:
- The authors have specifically worked with ETS fusion negative cell lines. Since a major proportion of cases show ETS fusions, further describing the effect of this combination in ERG/ETS positive cell lines (like VCaP) would be more powerful. While NCI-H660 is ERG positive, it is an NEPS with loss of AR activity.
Thank you for the interesting observation. We have evaluated a total of 14 cancer cell lines, including 7 PCa/CRPC/NEPC, and have yet to find inherent resistance to RCF + Ixz. Although we have not evaluated ERG/ETS positive VCaP, it would be surprising if they are inherently resistant. Further investigations will be required.
- While the synergistic effect of the drugs on cell death is evident in the current experiments, an immunohistochemical/ immunocytological demonstration of the targets of the drug and markers of cell apoptosis (c-PARP) would increase the strength of the study. In addition, demonstration of cytological vacuolation and the proportion of such cells for autophagy assessment is recommended.
Thank you for the suggestions. We previously had the good fortune of having a talented Research Pathologist (Teresita Reiner) for immunohistochemistry or immunocytochemistry. Due to funding limitations, she is no longer in the lab (since 2017). However, microscopic images of PC3 cells treated with CsA (10 mM) + Btz (10 nM) were obtained early in the project. This result (included as Supplementary Figure S4) showed that CsA + Btz had a drastic effect on cell morphology compared to CsA, Btz, and control treated cells. In the revised Results, section 3.1 (lines 239-241), we add “CsA + Btz had strong effects on PC3 cell morphology compared to CsA, Btz, and control cells (Supplementary Materials, Figure S4).” More detailed analysis of vacuolation and autophagy assessment is required.
We note that treatment of Hep3B (hepatocellular carcinoma) xenograft mice with RCF + Ixz results in increased vacuolated cells (proteotoxic stress) near blood vessels compared to vehicle, RCF, and Ixz (H&E staining of tumor sections). This data was presented as Figure S12 in our recently published manuscript https://doi.org/10.3390/ijms26146699.
- Since Autophagy is more recognized in NEPCs in comparison to AR+ CRPC / PC, please provide comparative evidence of cell death across AR+ and AR-lines. Do any of the cell lines have higher efficacy to this combination. Please provide figures and images to describe that.
Thank you for the suggestion. We have data that NEPC (H660, LASCPC, TRAMP C2) are more sensitive to RCF + Ixz compared to AR+ LNCaP and 22Rv1 cells. Other cells used were AR− PC3 (also sensitive) and DU145 (less sensitive). This data was presented as Supplementary Figure S2.
Round 2
Reviewer 1 Report
Comments and Suggestions for Authors
This study presents a compelling novel strategy to overcome the resistance of solid tumors to proteasome inhibitors. The combination of rencofilstat and ixazomib selectively induces apoptotic cell death in prostate cancer models while sparing non-cancer cells. The efficacy is mechanistically driven by a dual modulation of the unfolded protein response, uniquely sustaining the pro-death role of XBP1s while simultaneously inhibiting pro-survival PERK signaling. Furthermore, the combination disrupts the cyclophilin B/CD147 axis, attenuating downstream survival signals. These findings robustly support the clinical potential of this combination therapy for advanced prostate cancer with a favorable therapeutic index.
Author Response
Thank you for the positive review.
Reviewer 2 Report
Comments and Suggestions for Authors
Thank you for the updated manuscript and corresponding comments. We understand that research is time and resource dependent and it might not be possible to include every bit of detail within a single manuscript.
My only concern remains with the absence of true AR positive, ERG expressing cell lines which are representative of about 40-50% of human tumors in routine clinical practice. To provide a more realistic picture to the readers, I would suggest adding this message in the limitations or future directions.
Author Response
Thank you for the suggestion. In the revised Discussion (future studies, lines 504-505), we add “address in AR+/ERG+ PCa (represents 40-50% of routine clinical cases; [83]) cell line (e.g., VCaP) the effectiveness of RCF + Ixz.” We also add a new reference cited as #83 (http://doi.org.10.3389/fcell.2021.623809).